# Tailoring near-field thermal radiation between metallo-dielectric multilayers using coupled surface plasmon polaritons

Mikyung Lim [1], Jaeman Song [1], Seung S. Lee[1] & Bong Jae Lee [1]

Several experiments have shown a huge enhancement in thermal radiation over the black-body limit when two objects are separated by nanoscale gaps. Although those measurements only demonstrated enhanced radiation between homogeneous materials, theoretical studies now focus on controlling the near-field radiation by tuning surface polaritons supported in nanomaterials. Here, we experimentally demonstrate near-field thermal radiation between metallo-dielectric multilayers at nanoscale gaps. Significant enhancement in heat transfer is achieved due to the coupling of surface plasmon polaritons (SPPs) supported at multiple metal-dielectric interfaces. This enables the metallo-dielectric multilayers at a 160-nm vacuum gap to have the same heat transfer rate as that between semi-infinite metal surfaces separated by only 75 nm. We also demonstrate that near-field thermal radiation can be readily tuned by modifying the resonance condition of coupled SPPs. This study will provide a new direction for exploiting surface-polariton-mediated near-field thermal radiation between planar structures.

[1] Department of Mechanical Engineering, Korea Advanced Institute of Science and Technology, 291 Daehak-ro, Yuseong-gu, Daejeon 34141, South Korea. These authors contributed equally: Mikyung Lim, Jaeman Song. Correspondence and requests for materials should be addressed to S.S.L. (email: sslee97@kaist.ac.kr) or to B.J.L. (email: bongjae.lee@kaist.ac.kr)

Conventionally, thermal radiation emitted from real materials is limited by blackbody radiation, which can be described spectrally by Planck's law[1]. The blackbody radiation limit is, however, only applicable to the far-field radiation associated with propagating waves. When two bodies are placed at a sub-wavelength distance (i.e., in a near field), radiative heat transfer between them can overcome the blackbody limit by orders of magnitude[2–5]. This extraordinary phenomenon (called near-field thermal radiation) is due to additional energy transport by tunneling of evanescent waves in the near field. In particular, when materials support surface polaritons, the tunneling of evanescent waves is dominant at the resonance condition, such that quasi-monochromatic radiative heat transfer occurs[3,5].

Continuous efforts have been made to experimentally demonstrate enhanced radiative heat transfer at nanoscale[6–20]. Although experiments have not been able directly to show the monochromatic nature of spectral heat flux, several studies have revealed the role of surface polaritons in the enhancement of near-field thermal radiation[9–14,19,20]. For example, recent experimental works have reported that strong enhancements in heat transfer between polar dielectric materials like SiC[11] or SiO2[12,20] separated by vacuum gaps of 25–50 nm resulted from increased spectral heat flux at resonance condition of surface phonon polaritons (SPhPs) corresponding to the characteristic wavelength of thermal radiation at room temperature. However, the resonance frequency of SPhPs is lower than the bandgap frequency of thermophotovoltaic (TPV) cells[21–25], such that we cannot expect a highly efficient near-field TPV system that uses polar dielectric materials as an emitter. On the other hand, metals have plasma frequencies that are much higher than the characteristic frequencies of thermal radiation at room temperature; thus, no substantial enhancement in near-field thermal radiation due to coupling of surface plasmon polaritons (SPPs) in metals is expected under typical conditions[26]. Accordingly, the predetermined resonance condition of homogeneous materials limits the wide range application of near-field thermal radiation.

To overcome these issues, focus has now shifted to controlling of near-field thermal radiation via the introduction of nanomaterials[24,25,27–37]. In particular, metallo-dielectric (MD) multilayers have been extensively studied because mutual interactions of surface polaritons at multiple interfaces inside multilayers[24,25,35,36] provide exotic features including tuning of near-field thermal radiation. Given that such tuning capability is a pivotal issue in enhancing the performance of electricity-generation systems[24,25,37–40] and in thermal management[41,42], an experimental demonstration of the modulation of thermal radiation between MD multilayers is indispensable. Unfortunately, the control of near-field thermal radiation between MD multilayers and/or nanomaterials has not been experimentally achieved yet.

In this work, we report on an experimental demonstration of near-field thermal radiation between MD multilayers separated by submicron vacuum gap distances controlled with a custom-built-MEMS-device-integrated platform. Using temperature sensors and capacitance sensors integrated in the MEMS device, simultaneous measurements of heat flux and vacuum gap distance are achieved. Compared to that predicted for bulk Ti media, significantly enhanced near-field thermal radiation is measured with Ti/MgF2 MD multilayers. By simply changing the configuration of the MD multilayers, tuning of near-field thermal radiation is experimentally shown to be possible. The physical nature of the modulated near-field thermal radiation will be discussed, with exact calculation of the near-field thermal radiation between multilayered structures and analysis of the SPP resonance conditions. Further, comments on the validity of effective medium theory and the existence of a hyperbolic mode in the MD multilayered structure will be provided.

## Results

**Experimental setup.** To measure the near-field thermal radiation between MD multilayers, we fabricated a novel integrated platform consisting of MEMS-based microdevices and a three-axis nanopositioner (Fig. 1a). For MD multilayers, three Ti (metal)/MgF2 (dielectric) unit cells on bulk Ti substrates are employed. The thicknesses of the Ti and MgF2 layers are estimated by transmission electron microscope measurement to be 10 nm and 90 nm, respectively (Supplementary Figure 1). Accordingly, the volume filling ratio, defined as $f = t_m/(t_m + t_d)$, where $t_m$ is the thickness of the Ti layer and $t_d$ is the thickness of the MgF2 layer, is 0.1. The emitter part of the microdevice includes an MD emitter and a feedback-controlled heater (Fig. 1b, c). The receiver part of the microdevice contains an MD receiver and a temperature sensor (Fig. 1d). The width and the length of the emitter (i.e., 720 μm and 14.3 mm, respectively) were designed to be slightly larger than those of the receiver (i.e., 540 μm and 14.0 mm) so that the heat transfer area could be safely regarded as that of the receiver (i.e., the smaller one). The detailed fabrication process is described in Supplementary Note 1 and Supplementary Figure 2 and the cleanliness, the planarity, and the roughness of the sample surfaces are discussed in Supplementary Note 2 and Supplementary Figures 3-5.

A three-axis nanopositioner, which was composed of three picomotor actuators (8302-V, Newport) and three linear motion guides, was used for precise control of the vacuum gap distance between the MD emitter and the MD receiver. The moving stage of the nanopositioner that held the emitter part was placed on the three linear motion guides (Supplementary Note 3 and Supplementary Figure 6). Thus, each displacement made by each picomotor actuator changed the position and angle of the emitter part (refer to Fig. 1a, b), which allowed us to control the vacuum gap distance as well as the parallelism between the MD emitter and the MD receiver. To measure the vacuum gap distance as well as the curvature and the parallelism, the MD receiver is divided into four segments. The four local vacuum gaps are estimated from the measured capacitances between the MD-emitter-capacitor electrode (Fig. 1e) and each of the MD-receiver-capacitor electrodes (Fig. 1f). Detailed information on the structure of the MEMS-based microdevices and the data acquisition process is provided in Supplementary Note 4 and Supplementary Figures 7 and 8.

**Vacuum gap control and measurement.** The vacuum gap distance and the curvature (and parallelism) between the MD emitter and the MD receiver can be measured by dividing the MD receiver into four segments and measuring capacitances between the MD emitter and each of the MD receiver segments (see Fig. 2a and Supplementary Note 5). Four local vacuum gaps (denoted as $d_1$, $d_2$, $d_3$, and $d_4$) were sequentially measured and the vertical shift of each picomotor actuator was controlled to align the MD emitter in parallel with the MD receiver (i.e., $d_1 \approx d_4$ and $d_2 \approx d_3$). The curvature (and parallelism) between the MD emitter and the MD receiver can be evaluated using a fitted curve of the four local vacuum gaps, as shown in Fig. 2b. When the average gap distance $d = 200$ nm, the maximum curvature is estimated as 0.0029 m$^{-1}$, and the difference between $d_1$ and $d_4$ along the MD surfaces is only 4.7 nm (i.e., parallelism). Please note that the 'average' vacuum gap distance, $d$, between the MD emitter and the MD receiver is estimated via the Derjaguin approximation[43] using the four local gaps (see Supplementary Note 6 and Supplementary Figure 9 for details); thus, we can safely take account of any effect

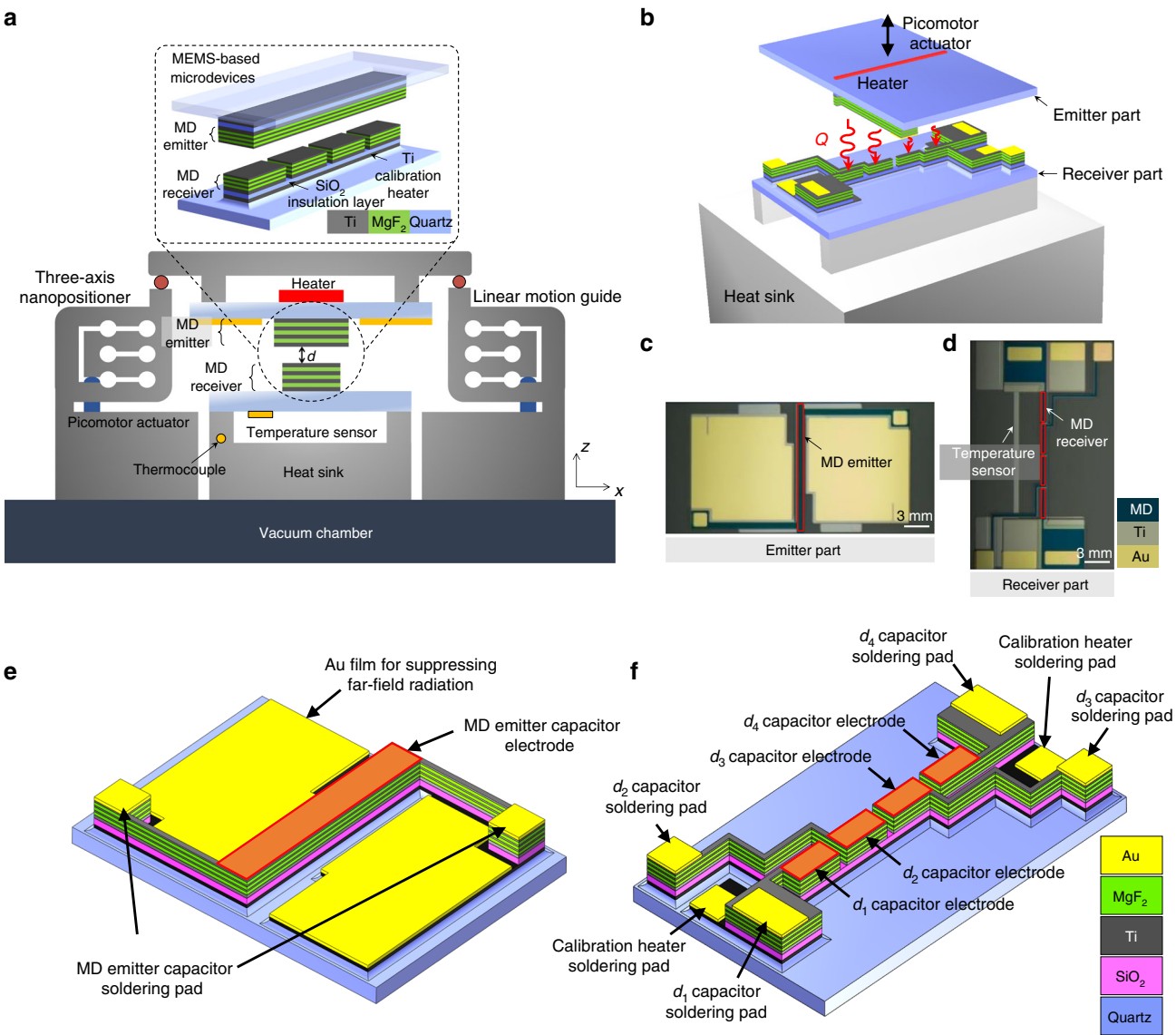

**Fig. 1** Experimental setup for measuring near-field thermal radiation between MD multilayers. **a** Schematic of an integrated platform consisting of the MEMS-based microdevices and the three-axis nanopositioner. **b** Three-dimensional schematic of the experimental setup. The position of the emitter part is controlled by the displacement of the picomotor actuators. The receiver part is fixed on the heat sink. **c** Photo of emitter part taken by digital single-lens reflex (DSLR) camera. The width and length of the MD emitter are 720 μm and 14.3 mm, respectively. The area excluding the MD emitter is coated with Au film to suppress far-field radiation. **d** DSLR image of the receiver part. The width of the MD receiver is 540 μm. The length of one segment of the MD receiver is 3.44 mm. **e** Three-dimensional schematic of emitter part of MEMS-fabricated microdevice. MD-emitter-capacitor electrode, Au film for suppressing far-field radiation, and MD-emitter-capacitor-soldering pads are described. **f** As in **e**, except for the receiver part. MD-receiver-capacitor electrodes ($d_{1-4}$ capacitor electrodes), MD-receiver-capacitor-soldering pads ($d_{1-4}$ capacitor-soldering pads), and calibration-heater-soldering pads are depicted

of bowing and tilting on radiative heat flux, as in previous studies[7,9,12,44]. Further, during the course of the measurement of near-field thermal radiation, it is imperative to ensure that there is no contact between the entire surfaces of the MD emitter and the MD receiver, because conduction heat transfer through physical contact can be falsely regarded as enhanced near-field thermal radiation. Contact between the MD emitter and the MD receiver can be detected by the signal of the dissipation factor, obtained by an LCR meter (E4980AL, Keysight). When the dissipation factor increases dramatically, the surfaces of the MD emitter and the MD receiver can be regarded as in contact (see Supplementary Figure 10 in Supplementary Note 7). For example, in Fig. 2c, when the dissipation factor for the fourth segment (noted as $D_4$)

rises dramatically, the average vacuum gap displacement between the MD emitter and the MD receiver increases abruptly; from these values, it can be considered that the two MD surfaces are in contact. Once contact is detected, the emitter is moved upward by the picomotor actuators until complete detachment between the emitter and the receiver is achieved. All measured data are obtained when there is no contact over the entire surfaces of the emitter and the receiver.

**Heat flux measurement**. In the measurement, the temperature of the MD emitter can be maintained at a designated value by feedback control of the input voltage to the heater (i.e., Joule heating) on the back of the emitter part (Fig. 3a). Conduction and

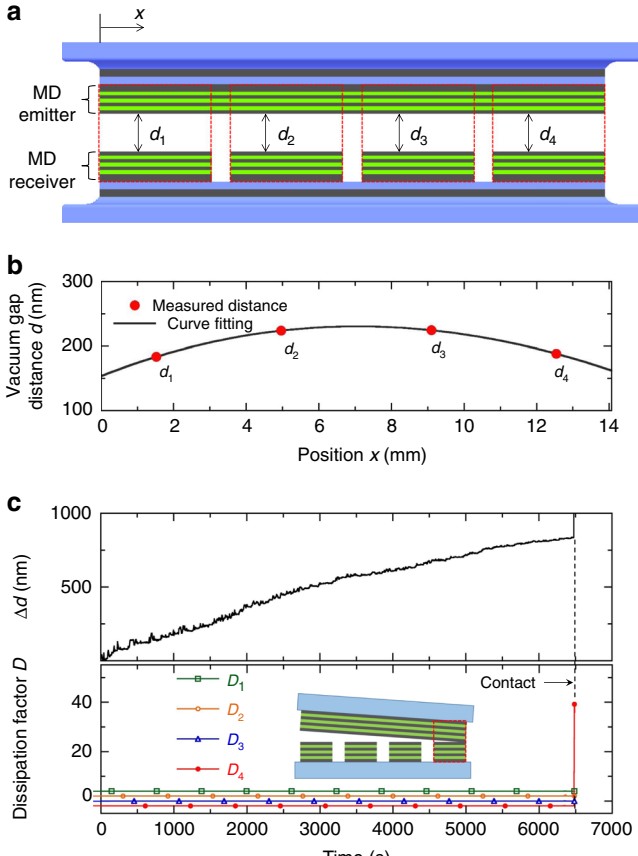

**Fig. 2** Measurement of vacuum gap distance between MD emitter and MD receiver. **a** Schematic cross-sectional view of aligned MD emitter and MD receiver. The four local vacuum gaps can be estimated from the measured capacitances between the MD emitter and each of the divided MD receiver segments. **b** Measurement of four local vacuum gaps and a fitted curve when $d = 200$ nm. **c** Upper panel: measured average vacuum gap displacement between MD emitter and MD receiver while reducing vacuum gap distance. In order to conduct steady-state measurement, the vacuum gap distance is slowly reduced by about 0.125 nm per second. Lower panel: measured dissipation factors for four segments, obtained simultaneously with results from the upper panel. The inset shows the scenario of physical contact between MD emitter and fourth MD receiver segment

convection heat transfer through air is suppressed as the entire MEMS-device-integrated platform is placed in a vacuum chamber ($<1 \times 10^{-3}$ Pa). Because the emitter part is physically connected to the vacuum chamber, which is considered a thermal reservoir (Fig. 1a), the conduction heat transfer from the MD emitter to the MD receiver through the vacuum chamber can be ignored. Far-field radiation between the emitter part and the receiver part, excluding the MD emitter and the MD receiver (denoted as $Q_{back1}$ and $Q_{back2}$), is further minimized by depositing an Au film on the emitter part, as shown in Figs. 1e and 3a. The radiative heat flux between the MD emitter and the MD receiver is measured via an Au temperature sensor placed on the back of the receiver part and a thermocouple glued to the heat sink. Figure 3a, b respectively depict an enlarged cross-sectional view of the experimental platform and an equivalent thermal circuit. The radiative heat flux from the MD emitter to the MD receiver (denoted as $Q_{e \rightarrow r}$) is further divided into two pathways so that $gQ_{e \rightarrow r}$, with $g$ being the geometrical factor, flows through the temperature sensor, the thermocouple, and the vacuum chamber. Although the

background radiation is suppressed by the Au coating, $Q_{back1}$ cannot be ignored. Accordingly, $Q_{e \rightarrow r}$ is expressed as the linear relationship of the temperature difference between the sensor ($T_{sen}$) and the thermocouple ($T_{TC}$): $gQ_{e \rightarrow r} + Q_{back1} = (T_{sen} - T_{TC})/R_{hs}$, resulting in $Q_{e \rightarrow r} = (T_{sen} - T_{TC})/(gR_{hs}) - Q_{back1}/g$. With the linear relationship of $Q$ and $T_{sen} - T_{TC}$ obtained from heat-flux calibration (Supplementary Notes 8 and 9 and Supplementary Figures 11 and 12), $Q_{e \rightarrow r}$ is estimated from the temperature difference between the sensor and the thermocouple while the emitter part is moved close to the receiver part after heat-flux calibration. Using the proposed MEMS-based microdevices, the vacuum gap width between the MD surfaces, as well as $Q_{e \rightarrow r}$, can be simultaneously measured.

**Experimental results**. The measured radiative heat flux between the MD emitter and the MD receiver is plotted in Fig. 4a. The filling ratio of MD multilayers was set to 0.1 (i.e., $t_m = 10$ nm and $t_d = 90$ nm). At $d = 160$ nm, the near-field radiative heat flux between the MD multilayers is measured as $\sim$7000 W m$^{-2}$, which is more than 100 times larger than the calculated far-field value and about seven times larger than the blackbody limit (992 W m$^{-2}$). To achieve such a dramatic enhancement between bulk Ti media, the two Ti surfaces should be separated by a 75-nm vacuum gap. Thus, difficulties in maintaining a sub-100-nm vacuum gap for considerable near-field enhancement can be avoided by employing MD multilayers. Please note that the near-field radiative heat flux between the Ti/MgF$_2$ MD multi-layers is compared only with that between bulk Ti media (i.e., metals) here because metal layers function as emitting layers. Although the near-field radiative heat flux between the bulk MgF$_2$ media is significant due to strong phonon absorption band of MgF$_2$ (refer to Fig. 4a), the role of MgF$_2$ in this configuration is nothing but the supporting layer (i.e., dielectric layer). As an evidence, analogous enhancement can be obtained even if the MgF$_2$ layers are replaced with lossless dielectric layers ($\varepsilon_d = 1.488$, see Supplementary Note 10 and Supplementary Figure 13 for details), which do not emit/absorb any radiation. The measurements in Fig. 4a were in good agreement with the theoretical prediction obtained from the exact calculations (see Methods).

Although the temperature of the MD emitter is maintained at 400 K during the measurements, the thin shaded region in Fig. 4a shows the calculated radiative heat flux between MD multilayers for an emitter temperature of $400 \pm 5$ K. It is clear that slight fluctuation in the emitter temperature does not result in significant change in the radiative heat transfer. Nevertheless, the measured radiative heat flux when the emitter temperature is varied by 30 K is plotted in Fig. 4b, and the measured near-field thermal radiation is clearly distinguishable. Because the permittivities of the materials are nearly independent of the temperature in the experimental conditions, if the measurements are represented as the radiative heat transfer coefficient $h_R$ (see inset of Fig. 4b), all data fall on similar theoretical graphs. This result also confirmed that our measurements were self-consistent.

In order to elucidate the mechanism of enhancement and modulation of the near-field thermal radiation, we now consider the effect of the configuration of MD multilayers on the radiative heat flux. For comparison, microdevices with MD multilayers of one Ti/MgF$_2$ unit cell ($f = 0.1$), MD multilayers of three Ti/MgF$_2$ unit cells ($f = 0.23$, $t_m = 25$ nm and $t_d = 85$ nm), and bulk Ti ($f = 1$) were additionally fabricated. In Fig. 4c, a considerable difference in radiative heat flux between MD multilayers is observed as the number of unit cells increases, meaning that the enhancement in near-field thermal radiation via introduction of MD multilayers is not just from the contributions of the first two or three interfaces but from those of multiple interfaces inside the

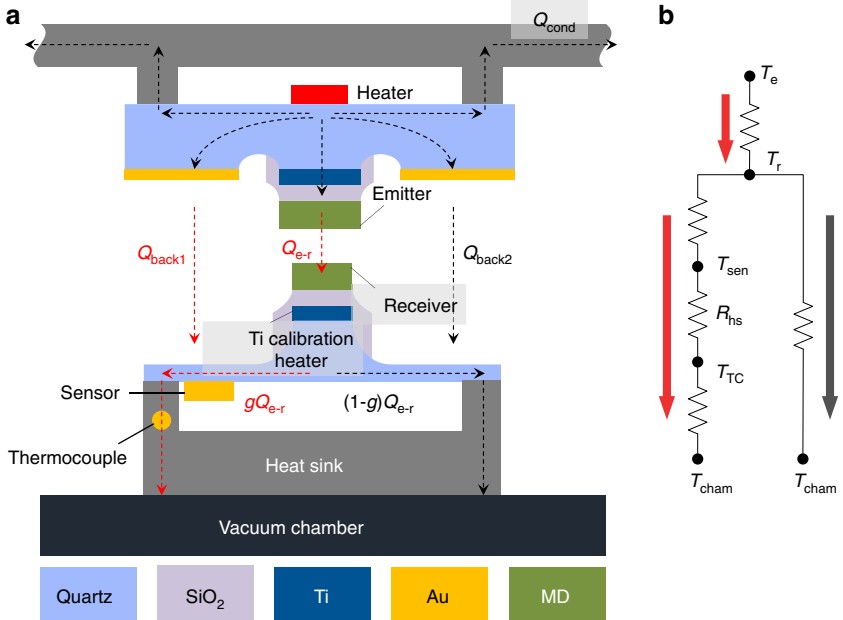

**Fig. 3** Heat transfer analysis with experimental setup. **a** Schematic of enlarged cross-sectional view of MEMS-device-integrated platform showing heat flow. The radiative heat flux from the MD emitter to the MD receiver $Q_{e \to r}$ can be estimated from the measured temperature difference between the backside temperature sensor and the thermocouple on the heat sink. **b** Equivalent thermal circuit for experimental platform

MD multilayers. Further, it is clearly shown in Fig. 4d that the MD multilayers with smaller filling ratio yield higher radiative heat transfer. The error bars shown in Fig. 4 are obtained following the procedure explained in Supplementary Note 11.

## Discussion

In Fig. 5a, the spectral heat flux between the bulk Ti media can be seen to have a peak at a low frequency, whereas the spectral heat flux between the MD multilayers of $f = 0.1$ (three Ti/MgF$_2$ unit cells) has a peak at angular frequency, $\omega$, of $\sim 1.2 \times 10^{14}$ rad s$^{-1}$. Interestingly, the spectral heat flux between MD multilayers of $f = 0.23$ (three Ti/MgF$_2$ unit cells) has peaks at both frequencies. When the spectral heat flux is plotted for each polarization ($p$-polarization for Fig. 5b and $s$-polarization for Fig. 5c), the enhancement mechanism is clearly revealed. The near-field thermal radiation between Ti media via $p$-polarization is extremely small due to the high plasma frequency of Ti so SPPs cannot play a substantial role in photon tunneling. However, when MD multilayers are introduced, broadband enhancement is clearly observed in the spectral heat flux for $p$-polarization. This enhancement is stronger for a smaller value of $f$. On the other hand, Fig. 5c suggests that the near-field thermal radiation between bulk Ti media is mainly via $s$-polarization[4]. The spectral heat flux for $s$-polarization is broadened because of the increased scattering in thin Ti film, but the magnitude of the peak is reduced as the filling ratio decreases.

In Fig. 5d–f, $S_{\beta,\omega}^{p}$ (see Methods for the definition) is plotted in log scale with SPP dispersion curves for each configuration[25,45,46]. Although $S_{\beta,\omega}^{p}$ becomes relatively large along the SPP dispersion curve close to the vacuum light line, for bulk Ti, such enhancement as occurs in the sharp $\beta$ region cannot contribute much to the total heat transfer. However, for the MD multilayers, a coupling of the SPPs supported at the Ti/MgF$_2$ interfaces within the thin Ti layer can result in splitting of the SPP modes, relocating the SPP branches to a larger $\beta$ region. Further, because of the repeated structures of MgF$_2$–Ti–MgF$_2$ in the MD multilayer, multiple SPP branches appear in the frequency region of interest.

Finally, a coupling of the SPPs from the MD emitter with those from the MD receiver is also observed ($\beta c_0 / \omega \sim 2$ and $\omega \sim 1.2 \times 10^{14}$ rad s$^{-1}$) and yields a substantial enhancement in the spectral heat flux. When comparing Fig. 5e with Fig. 5f, splitting of the SPP curves is more obvious in Fig. 5e; this splitting led to significant enhancement in the spectral heat flux when $f = 0.1$. To sum up, through analysis of the contour of $S_{\beta,\omega}^{p}$ together with the SPP dispersion curves, it is shown that the enhancement in near-field thermal radiation between MD multilayers originates mainly from coupling of SPPs supported in MD multilayers. This is experimentally verified in Fig. 4. As mentioned earlier, we employed Ti as emitting layers (i.e., metal layers) in MD multilayers and MgF$_2$ as supporting layers (i.e., dielectric layers), such that we can exploit the coupling of SPPs at multiple metal-dielectric interfaces. If we want to achieve the near-field enhancement over the bulk MgF$_2$ media, the same enhancement mechanism demonstrated in this work can be applied but MgF$_2$ needs to act as the emitting layer with another dielectric supporting layer. For example, MgF$_2$/dielectric ($\varepsilon_d = 1.35$) multilayers, where the thickness of both MgF$_2$ and dielectric layers is set to 200 nm, yield 21% enhanced near-field radiative heat flux than bulk MgF$_2$ media at vacuum gap of 200 nm. In Supplementary Note 12 and Supplementary Figure 14, it is shown that this enhancement is also through multiple interactions of SPhPs supported at MgF$_2$/dielectric interfaces. The same enhancement mechanism can be applied to other polar dielectric materials, such as SiO$_2$ and SiC[36].

The Ti/MgF$_2$ multilayered structure discussed here can also be described with effective medium theory (EMT), given that the periodicity ($\Lambda = 100$ nm) is much smaller than the characteristic wavelength of thermal radiation under experimental conditions[29–31,36]. Computed parallel ($x$) and perpendicular ($z$) components of uniaxial effective permittivities[29–31,36,47] for MD multilayers with a filling ratio of 0.1 show that a hyperbolic band (type II, $\varepsilon_x < 0$, $\varepsilon_z > 0$) is extended over most of the frequency range of interest. This broad hyperbolic band ranges from $1.2 \times 10^{14}$ rad s$^{-1}$ to $9.2 \times 10^{14}$ rad s$^{-1}$, which is the frequency interval

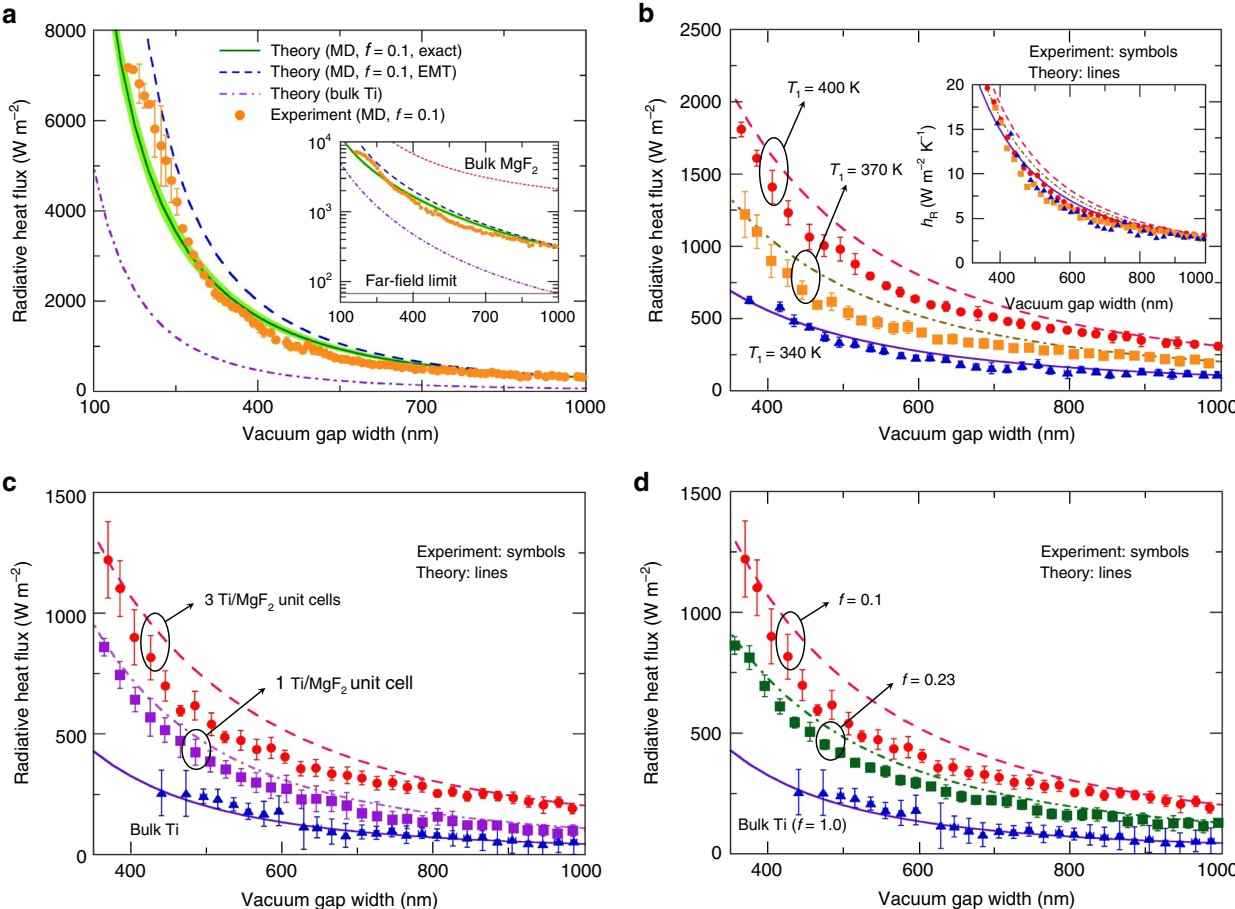

**Fig. 4** Manipulation of near-field radiation by modifying surface condition with MD multilayers. **a** Measured near-field radiative heat flux between MD multilayers ($f = 0.1$, 3 Ti/MgF$_2$ unit cells) with respect to the submicron vacuum gap distance (emitter temperature: 400 K and receiver temperature: 300 K). Theoretical results of near-field thermal radiative heat flux between multilayered structures that are computed considering multiple reflections in multilayer (i.e., exact computation) or effective medium theory (EMT), as well as between bulk Ti media, are plotted. The inset shows the near-field radiative heat flux plotted on log scale. Calculated near-field radiative heat flux between bulk MgF$_2$ media and far-field radiative heat flux between Ti/MgF$_2$ multilayers are also plotted. **b** Near-field radiative heat flux between MD multilayers ($f = 0.1$, 3 Ti/MgF$_2$ unit cells) as a function of emitter temperature. The inset is a graph of radiative heat transfer coefficient $h_R$ for all measured data. **c** Enhanced near-field radiative heat flux between MD multilayers ($f = 0.1$) with different numbers of unit cells. Emitter temperature is set to 370 K. **d** Tuning of near-field radiative heat flux by changing of volume filling ratio of MD multilayer, $f$ (3 Ti/MgF$_2$ unit cells). The near-field radiative heat flux is measured at an emitter temperature of 370 K. The error bars in **a**–**d** represent the combination of the measurement uncertainty and standard deviation of multiple measurements (Supplementary Note 11)

where coupled SPPs are observed. In Fig. 5, it can easily be seen that the coupled SPPs of bulk Ti media persist for the MD multilayer structures[35]. This "surface mode", associated with coupling of the topmost Ti layers of the MD multilayers, can be distinguished from the mode of coupled SPPs at the multiple interfaces within the MD multilayers. We have checked the Bloch wave dispersion by assuming infinite Ti/MgF$_2$ multilayers and found that coupled SPPs in the repeated MgF$_2$–Ti–MgF$_2$ structure inside the MD multilayer, as shown in Fig. 5e, f, are inside the hyperbolic Bloch bands as predicted in previous studies;[29,30,33,35] that is, these high-$k$ waves (hyperbolic mode) are propagating inside the MD multilayer. Thus, the significant broadband enhancement shown in Fig. 5b is mainly due to hyperbolic mode, which enables MD multilayers to have hyperbolic-metamaterial-like properties.

The near-field radiative heat flux between MD multilayers can also be calculated using EMT[29,31]. In general, it is known that EMT can be safely applied when the vacuum gap distance is much larger than the unit cell period of MD multilayers[29,31]; however, because EMT does not take account of the coupled SPPs

supported at the topmost layers of each MD structures, EMT can be safely applied if the effect of the surface mode of the topmost layer is reduced[30,35]. As can be seen in Fig. 4a, EMT overestimates the heat transfer by more than 50% when $d \leq 200$ nm (i.e., two times the period) and EMT provides radiative heat flux with a deviation of less than 10% when $d \geq 600$ nm. In contrast to the case in which the topmost layer is Ti, if the topmost layer of the MD multilayer is MgF$_2$, the EMT converges to the exact calculation at larger gaps; this is due to strong coupling of SPhPs supported at the topmost MgF$_2$ layers across the large vacuum gaps.

As discussed throughout this paper, a significant enhancement of near-field thermal radiation can be achieved by having contributions from multiple metal/dielectric interfaces[36]. However, it has also been suggested that an enhancement of near-field thermal radiation similar to that obtained using an MD multilayer can be achieved with a thin metal film[48]. For our Ti/MgF$_2$ MD multilayer, either a thin film or a multilayer structure can be beneficial in enhancing heat transfer, depending on the configuration and vacuum gap width. Nevertheless, it is worthwhile to

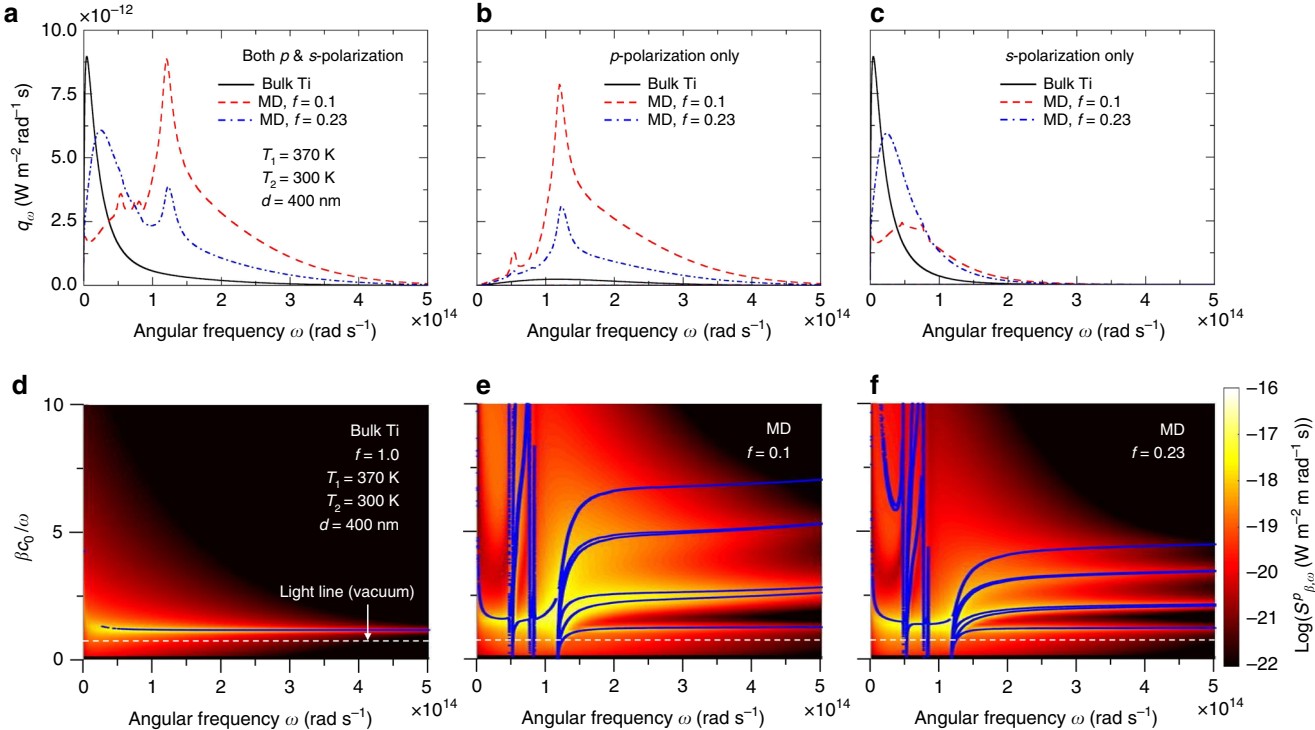

**Fig. 5** Investigation of manipulated near-field radiative heat flux by modifying surface condition with MD multilayers. **a** Computed spectral heat flux between Ti/MgF$_2$ multilayers ($f = 0.1$ and 0.23) and bulk Ti media ($f = 1$) at vacuum gap distance of 400 nm. The temperatures of the emitter and receiver are set to 370 K and 300 K, respectively. **b, c** As in **a**, but for $p$-polarized and $s$-polarized spectral heat flux, respectively. **d** Computed $S_{\beta,\omega}^{p}$ between bulk Ti media for vacuum gap distance of 400 nm and temperature of emitter and receiver conditions described in **a**. A larger value of $S_{\beta,\omega}^{p}$ is observed along the plotted SPP dispersion curve. **e, f** As in **d**, but between MD multilayers, which have volume filling ratios of 0.1 and 0.23

mention that the advantage of an MD multilayer is not just the large heat flux enhancement but the physical nature of such enhancement[35]. That is, for Ti thin films, surface modes, which are strongly confined to the surface, dominate the heat transfer, whereas hyperbolic modes, which propagate inside the MD multilayer, are dominant for MD multilayers, making MD multilayers advantageous for thermal management applications or near-field TPV applications[35].

We developed a novel integrated platform for measuring the near-field thermal radiation between MD multilayers (made of alternating Ti and MgF$_2$ layers) with an active heat transfer area of 7.56 mm². We demonstrate that the near-field thermal radiation can be tuned using coupled SPPs supported in an MD multilayered structure and that, due to multiple interactions of SPPs supported in each metal/dielectric interface, the corresponding radiative heat transfer is larger than that between metals. As a result, compared to the calculated far-field case, the near-field radiation between MD multilayers at $d = 160$ nm was enhanced ~100-fold and was equivalent to that between bulk Ti media at $d = 75$ nm. Our findings are quite significant because experiments looking at near-field thermal radiation between planar geometries separated by tens of nanometers have been conducted with only extremely small heat transfer area (~0.0025 mm²)[11,12,20]. Although the measured heat transfer rate between MD multilayers is not the largest value among those of all homogeneous materials, the enhancement mechanism shown here can be applied to other configurations, including polar dielectric materials, to achieve even better values of heat transfer rate. Further, the tailoring of near-field thermal radiation achieved by coupled SPPs supported in metal/dielectric interfaces has been reported to make much more efficient near-field TPV systems[24,25,37]. Thus, the results obtained in this study will pave

the way for real-world applications of near-field radiation transferred by MD multilayers.

## Methods

**Near-field thermal radiation between MD multilayers.** The near-field thermal radiation between MD multilayers was calculated as:

$$q''_{\text{net}} = \int_0^\infty d\omega \, q''_{\omega,\text{net}} = \int_0^\infty d\omega \int_0^\infty \left[ S_{\beta,\omega}^{p}(\beta,\omega) + S_{\beta,\omega}^{s}(\beta,\omega) \right] d\beta, \quad (1)$$

where $\omega$ is the angular frequency and $\beta$ is the parallel wavevector component. In the above equation, $S_{\beta,\omega}^{p,s}(\beta,\omega)$ is derived for propagating (i.e., $\beta < \omega/c_0$) and for evanescent (i.e., $\beta > \omega/c_0$) waves in vacuum[5]:

$$S_{\beta,\omega,\text{prop}}^{p,s}(\beta,\omega) = \frac{\Theta(\omega,T_1) - \Theta(\omega,T_2)}{\pi^2} \times \frac{\beta(1 - |r_{01}^{p,s}|^2)(1 - |r_{02}^{p,s}|^2)}{4|1 - r_{01}^{p,s} r_{02}^{p,s} e^{i2k_{0z}d}|^2}$$

$$S_{\beta,\omega,\text{evan}}^{p,s}(\beta,\omega) = \frac{\Theta(\omega,T_1) - \Theta(\omega,T_2)}{\pi^2} \times \frac{\beta \text{Im}(r_{01}^{p,s}) \text{Im}(r_{02}^{p,s}) e^{-2\text{Im}(k_{0z})d}}{|1 - r_{01}^{p,s} r_{02}^{p,s} e^{i2k_{0z}d}|^2}, \quad (2)$$

where Im() takes the imaginary part of a complex quantity and $k_{0z}$ is the normal wavevector component in vacuum. $T_1$ and $T_2$ are the temperatures of the emitter (medium 1) and the receiver (medium 2) and $\Theta(\omega,T_i) = \frac{\hbar\omega}{\exp\{\hbar\omega/(k_B T_i)\} - 1}$ is the mean energy of the Planck oscillator with $\hbar$ representing the Planck constant divided by $2\pi$ and $k_B$ of the Boltzmann constant. Here, $r_{ij}^{p,s}$ is the modified reflection coefficient at the $i$–$j$ interface (vacuum: medium 0), which was derived from Airy's reflection formulas for a multilayer[49]. The optical properties of Ti were obtained from ref. [50] and the additional electron boundary scattering at the interface was considered as in refs. [51,52] because the thickness of the Ti film is smaller than its electron mean free path. The dielectric functions of MgF$_2$ and SiO$_2$ were taken from ref. [53]. Although the SiO$_2$ insulator and Ti calibration heater were placed below the MD receiver, overall structure could safely be considered a multilayered structure on bulk Ti substrate because a thick Ti layer (180 nm) was deposited between the Ti/MgF$_2$ multilayered structure and the SiO$_2$ insulating layer (see Supplementary Note 13 and Supplementary Figure 15).

## Data availability

The data that support the findings of this study are available from the corresponding authors upon reasonable request.

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

## Acknowledgements

This research was supported by the Basic Science Research Program through the National Research Foundation of Korea (NRF) funded by the Ministry of Science, ICT and future Planning (NRF-2017R1A2B2011192 and NRF-2015R1A2A1A10055060).

## Author contributions

All authors participated in conceiving the experiment. The MEMS-based microdevices were designed and fabricated by M.L. and J.S. under the supervision of S.S.L. The three-axis nanopositioner was designed and fabricated by J.S. under the supervision of B.J.L. The experimental data were obtained by M.L. and J.S. The data analysis was conducted

by M.L., J.S., and B.J.L. Theoretical modeling was performed by M.L. All authors contributed to the writing of the manuscript.

## Additional information

**Competing interests:** The authors declare no competing interests.

