## [Peer Review File · Nature Communications]

Reviewers' Comments:

Reviewer #1:

Remarks to the Author:

The authors have largely addressed the detailed questions I raised in my previous review. Specifically, the description of the experimental platform is much more improved in the revised manuscript and the SI. Overall, I think this is a commendable advancement and therefore I recommend publication after the following minor concerns are addressed.

1) In Fig. S3, I recommend adding data that show the planarity along the length of the devices (which was my original concern). In my opinion, this will help greatly in adding more confidence in the results.

2) There are a few recent advances in the field that should probably be cited:

a) The discussion in paragraph 2 probably needs a revision as recent work (A. Fiorino et al. Nano Letters (2018)) has achieved gaps of the order of 20-30 nms and showed an ~ 1000 fold enhancement in heat fluxes in heat transfer, which is significantly (~ 10 fold) larger than that reported in refs. 11 and 12 currently cited.

b) Recent work (M. Ghashami et al., Phys. Rev. Lett. 2018.) has explored near-field radiation between macroscopic planar surfaces using a nano-positioning platform and should probably be cited.

c) Finally, a near-field TPV system, which forms part of the motivation for the current work, has recently been experimentally studied (A. Fiorino et al., Nature Nanotechnology, 2018). This work should probably be cited.

Reviewer #2:

Remarks to the Author:

The reviewer would like to thank the authors for addressing the comments. Although the authors changed their focus from hyperbolic metamaterials to multilayers in the revised manuscript, the overall work is still incomplete and not convincing for publication in Nature Communications. As illustrated in Ref. 46 (Miller et al., Phys. Rev. Lett. 112, 157402, 2014), the near field enhancement between multilayers can be achieved by single layer films. In this case, a benchmark experiment with single layer films is quite necessary in order to fully demonstrate the functionality of multilayers. The authors mentioned in the response letter that one of the goals of this work is to show that the radiative transfer between multilayers is "larger than that between homogeneous materials". Clearly, they should not just compare with the case for semi-infinite metals, and an experiment with semi-infinite dielectrics should also be included.

Reviewer #3:

Remarks to the Author:

The authors have addressed all the comments of the reviewers and revised the manuscript accordingly. They have provided the detailed descriptions on the structure of the MEMS-based devices and the procedure of the gap estimation, which now strengthen the experimental results. From my point of view, the paper is almost suitable for publication in Nature Communications. However, I recommend the authors to address a following minor comment.

As for calibration of heat flux (in Supplementary Note 9), the authors varied the applied electric power to the Ti calibration heater Q and measured the temperature difference between the sensor

and thermocouple ($T_{sen}-T_{sc}$). My concern is whether the heat generation in the Ti calibration heater is equivalent to the increase of $Q_{e \rightarrow r}$. Since the position of the heat generation in the Ti calibration heater (relatively uniform inside the device) and that in the near-field experiment (localized at the surface) is different, the geometry factor g in both cases might be different.

Authors' Response

Manuscript ID: NCOMMS-18-17802-T

Title: Tailoring near-field thermal radiation between metallo-dielectric multilayers using coupled surface plasmon polaritons

Authors: M. Lim, J. Song, S. S. Lee, and B. J. Lee

We thank again the reviewers for his/her time and are also very thankful for giving us an opportunity to address concerns raised by the reviewers. We have carefully considered reviewers' comments and incorporated most of them to further improve the quality of our manuscript. Specifically, we have addressed all minor concerns raised by Reviewer 1 and Reviewer 3 and we conducted additional experiment following the Reviewer 2's suggestion. Detailed explanations along with the revised manuscript where changes are highlighted are as below.

[REVIEWER 1]

The authors have largely addressed the detailed questions I raised in my previous review. Specifically, the description of the experimental platform is much more improved in the revised manuscript and the SI. Overall, I think this is a commendable advancement and therefore I recommend publication after the following minor concerns are addressed.

1) In Fig. S3, I recommend adding data that show the planarity along the length of the devices (which was my original concern). In my opinion, this will help greatly in adding more confidence in the results.

[AR]: We appreciate the suggestion. Because the length of both the metallo-dielectric (MD) emitter and the MD receiver is 14.3 mm, we could not scan the MD-multilayer surfaces along the length via a confocal optical profiler (μ -surf, Nanofocus). As a substitute, a stylus line profiler (Dektak-8, Veeco) was used to obtain the line profile and confirm the planarity of the MD-multilayer surfaces along the length of the devices. Figure AR1a,b respectively indicate the scanning result of the MD-emitter surface and MD-receiver surface with respect to the length direction. As illustrated in Fig. AR1a, the MD-emitter surface is found to be concave-shaped slightly, and the largest deviation from the planarity was estimated as about 210 nm. We infer that intrinsic bowing of the fabricated device is one of the reason of the curvature shown in the Fig. 2b of the manuscript. The MD receiver was found to have less bowing than the MD emitter (see Fig. AR1b), and the largest deviation from the planarity was estimated as about 110 nm. In addition to the intrinsic bowing of the fabricated emitter and receiver parts, thermal expansion by heating as well as constraint of the devices to the three-axis nanopositioner could be factors of bowing of the MD-multilayer surfaces. Because we measured the four local gaps from the capacitance and obtained the average vacuum gap distance using the Derjaguin approximation (see Supplementary note 6), we have already considered the effect of the curved MD-multilayer surfaces on the near-field radiative heat flux.

Following the reviewer's suggestion, we have added figures and following discussion to Supplementary Note 2: *"In addition to the line profile over the width of the active region, a line profile along the length of the device is obtained with the stylus line profiler (see Figs. S4a and S4b). As illustrated in Fig. S4a, the MD-emitter surface is curved as a concave shape: the largest difference of deviation from planarity is about 210 nm; that of the MD receiver is estimated at about 110 nm (see Fig. S4b). Because the four local gaps are measured from the capacitance and the average vacuum gap distance is obtained using the Derjaguin approximation (see Supplementary note 6), the effect of the curved MD-multilayer surfaces on the near-field radiative heat flux is fully considered."*

[Figure AR1; newly added as Figure S4 of the revised Supplementary Notes]: Stylus line profiler images of the fabricated MD-multilayer surfaces along the length direction of the devices. (a) Emitter part, (b) Receiver part.

2) There are a few recent advances in the field that should probably be cited:

a) The discussion in paragraph 2 probably needs a revision as recent work (A. Firorino et al. Nano Letters (2018)) has achieved gaps of the order of 20-30 nms and showed an ~ 1000 fold enhancement in heat fluxes in heat transfer, which is significantly (~ 10 fold) larger than that reported in refs. 11 and 12 currently cited.

b) Recent work (M. Ghashami et al., Phys. Rev. Lett. 2018.) has explored near-field radiation between macroscopic planar surfaces using a nano-positioning platform and should probably be cited.

c) Finally, a near-field TPV system, which forms part of the motivation for the current work, has recently been experimentally studied (A. Fiorino et al., Nature Nanotechnology, 2018). This work should probably be cited.

[AR]: Thank you very much for your considerate comments. We have cited aforementioned references in the introduction.

[REVIEWER 2]

The reviewer would like to thank the authors for addressing the comments. Although the authors changed their focus from hyperbolic metamaterials to multilayers in the revised manuscript, the overall work is still incomplete and not convincing for publication in Nature Communications. As illustrated in Ref. 46 (Miller et al., Phys. Rev. Lett. 112, 157402, 2014), the near field enhancement between multilayers can be achieved by single layer films. In this case, a benchmark experiment with single layer films is quite necessary in order to fully demonstrate the functionality of multilayers. The authors mentioned in the response letter that one of the goals of this work is to show that the radiative transfer between multilayers is “larger than that between homogeneous material”. Clearly, they should not

just compare with the case for semi-infinite metals, and an experiment with semi-infinite dielectrics should also be included.

[AR]: As discussed throughout the paper, a significant enhancement of near-field thermal radiation can be achieved by having contributions from multiple metal/dielectric interfaces [Iizuka and Fan, Phys. Rev. Lett. 120, 063901, 2018]. However, it has also been suggested that an enhancement of near-field thermal radiation similar to that obtained using a metallo-dielectric (MD) multilayer can be achieved with a thin metal film [Miller et al., Phys. Rev. Lett. 112, 157402, 2014]. For our Ti/MgF₂ MD multilayer, either a thin film or a multilayer structure can be beneficial in enhancing heat transfer, depending on the configuration and vacuum gap width (see pages 4–6 of the original Author Response Letter). Therefore, we strongly believe that the benchmark experiment with MD multilayers of single Ti/MgF₂ unit cell on bulk Ti substrate is more appropriate to demonstrate the functionality of multilayers than that with the single layer films (i.e., thin Ti layer on MgF₂ substrate) for following reason.

In this manuscript, we have addressed three functionalities of the multilayers: (1) We have experimentally shown that the near-field thermal radiation can be significantly enhanced by introducing MD multilayers instead of bulk metals (see Fig. 4a). We also have theoretically shown that this enhancement results from the coupled SPPs supported at the **multiple interfaces in the MD multilayers**; (2) We have demonstrated that the near-field thermal radiation can be controlled by modifying the resonance conditions of the coupled SPPs which is achieved by **changing the configuration of the MD multilayers** (see Fig. 4c of original manuscript); and (3) As discussed in page 11 of the original manuscript, the coupled SPPs supported at the **repeated dielectric-metal-dielectric structure inside MD multilayers** are in the hyperbolic Bloch bands which enables MD multilayers to have hyperbolic-metamaterial-like properties.

It should be noted that for single layer film, surface modes, which are strongly confined to the vacuum-metal interface, dominate the heat transfer, whereas for the MD multilayers hyperbolic modes, which

[Figure AR2; newly added as Figure 4c of the revised manuscript]: Benchmark experimental results obtained with the sample of single Ti/MgF₂ unit cell on a bulk Ti substrate. The original experimental data (three Ti/MgF₂ unit cells on a bulk Ti substrate) are compared with the benchmark experimental results.

propagate inside the MD multilayers, are dominant. Therefore, in order to fully demonstrate the functionality of multilayers (i.e., coupled SPPs at multiple interfaces), we conducted new experiment with the MD multilayers of single Ti/MgF₂ unit cell and corresponding results were added in the revised manuscript (see Fig. AR2 as well as Fig. 4c of the revised manuscript). With this newly-conducted experiment, we have successfully shown that the significant enhancement can be achieved through the coupled SPPs supported at the multiple interfaces in the MD multilayers; that is, the functionality of the multilayer is now experimentally confirmed. It is clearly demonstrated that such enhancement is not just from the contribution of top surfaces, which was original concern of the Reviewer 2, but from the contributions of the multiple interfaces inside the MD multilayers.

In the original response letter, we have clearly written that our goal is to show that the radiative heat transfer between multilayers is “larger than that between homogenous materials (i.e., semi-infinite metal surfaces)” due to multiple interactions of SPPs supported in each interfaces (please refer to page 18 of the original Author’s Response Letter). Further, throughout the manuscript, we have continuously written (from abstract to conclusion) that the near-field thermal radiation between MD multilayers is larger than that between “metals”.

As we thoroughly discussed, we compare the near-field thermal radiation between Ti/MgF₂ MD multilayers with the case between bulk Ti media because Ti is the main emitter in the near-field thermal radiation between Ti/MgF₂ MD multilayers. In this case, MgF₂ acts as nothing but a supporting layer, and replacing MgF₂ with a lossless dielectric layer does not change anything fundamentally (see page 7 of the original Author’s Response Letter). It is quite obvious that the radiative heat transfer between lossless dielectric media is impossible to be measured because they do not emit and absorb radiation. The reviewer’s claim about the experiment with semi-infinite dielectrics will not produce any meaningful outcome.

[REVIEWER 3]

The authors have addressed all the comments of the reviewers and revised the manuscript accordingly. They have provided the detailed descriptions on the structure of the MEMS-based devices and the procedure of the gap estimation, which now strengthen the experimental results. From my point of view, the paper is almost suitable for publication in Nature Communications. However, I recommend the authors to address a following minor comment.

1) As for calibration of heat flux (in Supplementary Note 9), the authors varied the applied electric power to the Ti calibration heater Q and measured the temperature difference between the sensor and thermocouple ($T_{\text{sen}} - T_{\text{TC}}$). My concern is whether the heat generation in the Ti calibration heater is equivalent to the increase of $Q_{\text{e} \rightarrow \text{r}}$. Since the position of the heat generation in the Ti calibration heater (relatively uniform inside the device) and that in the near-field experiment (localized at the surface) is different, the geometry factor g in both cases might be different.

[AR]: We understand reviewer’s concern. Although the Ti calibration heater and the top receiver surface seem to be separated considerably in Fig. 3, the distance between Ti calibration heater and the top surface is actually about 2.5 μm which is much shorter than the thickness of the receiver part (i.e., 1 mm). In other words, the thermal resistance of the heat path between Ti calibration heater and the top surface can be neglected. Figure AR3 shows the COMSOL Multiphysics[®] simulation results when there is no background radiation. As can be seen in the real-scale description of the receiver part on the heat sink, the position of Ti calibration heater and the top receiver surface is almost indistinguishable. The simulation results also show that the change of $T_{\text{sen}} - T_{\text{TC}}$ are

almost same for heat generation in the Ti calibration heater and heat flux through top receiver surface.

In the revised Supplementary Note 9, we have added following sentence to backup the validity of heat-flux calibration: *“Please note that the thermal resistance of the heat path between the Ti calibration heater and the top surface is negligible compared to that of the heat path between the Ti calibration heater and the temperature sensor, meaning that geometrical factor g is almost constant for heat-flux calibration condition and near-field experiment condition. Thus, the relationship between Q and $T_{sen} - T_{TC}$ can be safely used to estimate $Q_{e \rightarrow r}$ from the measured $T_{sen} - T_{TC}$ value.”*

[Figure AR3]: (a) Real-scale description of the receiver part on the heat sink. (b) COMSOL Multiphysics[®] simulation results: the temperature difference between the sensor and thermocouple ($T_{sen} - T_{TC}$) under the heat flux condition (i.e., near-field experiment condition) and the heat generation condition (i.e., heat-flux calibration condition).

Reviewers' Comments:

Reviewer #1:

Remarks to the Author:

The referees have addressed all my questions and I recommend publication.

Reviewer #2:

Remarks to the Author:

(1) Although the authors showed that the near-field heat transfer between 3 Ti/MgF₂ unit cells are larger than that of 1 Ti/MgF₂ unit cell, this is not adequate and convincing result. In their experiment, clearly, the thickness of the Ti thin film in the case of 1 Ti/MgF₂ unit cell is not optimized, and therefore heat transfer is not maximized, which certainly can lead to a smaller heat transfer. A fundamental question to be answered is whether a multilayer is truly better than a single thin film for near-field heat transfer. If a single thin Ti film can achieve the same or even larger heat transfer, why does one bother to use a more complicated Ti/MgF₂ multilayer structure?

(2) For the second comment, please note that MgF₂ is not a lossless dielectric for infrared light. Instead it has strong infrared absorption in the middle infrared range (wavelength > 6 μm), which can result in high near-field heat transfer enhancement as well. As mentioned by the authors, they intended to show that the radiative heat transfer between multilayers is "larger than that between homogeneous materials (i.e., semi-infinite metal surfaces)". The Ti/MgF₂ multilayer structure that they used is made from both Ti and MgF₂. Here they only selected the experimental results in favor of their conclusion by intentionally referring "homogeneous materials" to "semi-infinite metal (Ti) surfaces" but ignoring the case of "semi-infinite dielectric (MgF₂) surfaces", which again is very misleading.

Reviewer #3:

Remarks to the Author:

In the revised version of the manuscript, the authors have fully addressed the comments of the three reviewers. The major claim of the paper (the enhancement of near-field thermal radiation with metallo-dielectric (MD) multilayers) has been more clearly supported by their additional experiment with the single MD structure. From my point of view, the paper is now suitable for publication in Nature Communications.

Authors' Response

Manuscript ID: NCOMMS-18-17802A

Title: Tailoring near-field thermal radiation between metallo-dielectric multilayers using coupled surface plasmon polaritons

Authors: M. Lim, J. Song, S. S. Lee, and B. J. Lee

We have carefully considered the Reviewer2's comments and incorporated them into the revised manuscript for helping general readers' understanding. Detailed explanations are given in the following.

[REVIEWER 1]

The referees have addressed all my questions and I recommend publication.

[AR]: We are grateful that the reviewer is now fully convinced by our response.

[REVIEWER 2]

(1) Although the authors showed that the near-field heat transfer between 3 Ti/MgF₂ unit cells are larger than that of 1 Ti/MgF₂ unit cell, this is not adequate and convincing result. In their experiment, clearly, the thickness of the Ti thin film in the case of 1 Ti/MgF₂ unit cell is not optimized, and therefore heat transfer is not maximized, which certainly can lead to a smaller heat transfer. A fundamental question to be answered is whether a multilayer is truly better than a single thin film for near-field heat transfer. If a single thin Ti film can achieve the same or even larger heat transfer, why does one bother to use a more complicated Ti/MgF₂ multilayer structure?

[AR]: First of all, we respectively disagree with the reviewer's argument and strongly believe that our additional experiment provides adequate and convincing result to demonstrate the enhancement of near-field thermal radiation with metallo-dielectric (MD) multilayers via coupling of surface plasmon polaritons (SPPs), which is the major claim of our manuscript.

Initially, the Reviewer 1 raised the same concern, and we have fully addressed this issue already (refer to pages 4–6 of the original Author Response). As we have continuously mentioned, the answer to reviewer's question can be found in the recent publication [Iizuka and Fan, Phys. Rev. Lett. 120, 063901, 2018], in which the authors stated that *"We point out that under the right condition, intermediate layers can facilitate the energy transfer from surface states located away from the middle vacuum gap. As a result, one can in fact achieve significant enhancement of near-field heat transfer by having contributions from multiple interfaces."*

Further, the reviewer insisted that the single unit cell results in smaller heat transfer than the multilayer because the thickness of the Ti thin film is not optimized. However, it can be also rephrased as: either a single thin film or a multilayer structure can be beneficial in enhancing heat transfer, depending on the configuration and vacuum gap width (we have clearly stated this on page 11 of the original manuscript). It should be noted that such configuration- or vacuum-gap-dependent near-field enhancement is originated from the coupling nature of SPPs supported at each of interfaces within multilayer structure. As clearly written in the introduction of our original manuscript, tuning capability of the near-field thermal radiation is a pivotal issue in enhancing the performance of electricity-generation systems and in thermal management. In this work, we have experimentally demonstrated new tuning

capability of the near-field thermal radiation by introducing MD multilayers for the wide range application. Please do not underestimate the importance of our finding by simply focusing on achieving the maximum heat transfer rate rather than appreciating the fundamental enhancement mechanism.

(2) For the second comment, please note that MgF_2 is not a lossless dielectric for infrared light. Instead it has strong infrared absorption in the middle infrared range (wavelength $> 6 \mu\text{m}$), which can result in high near-field heat transfer enhancement as well. As mentioned by the authors, they intended to show that the radiative heat transfer between multilayers is “larger than that between homogeneous materials (i.e., semi-infinite metal surfaces)”. The Ti/ MgF_2 multilayer structure that they used is made from both Ti and MgF_2 . Here they only selected the experimental results in favor of their conclusion by intentionally referring “homogeneous materials” to “semi-infinite metal (Ti) surfaces” but ignoring the case of “semi-infinite dielectric (MgF_2) surfaces”, which again is very misleading.

[AR]: Again, the Reviewer 1 initially raised the same concern, and we have addressed this issue already (refer to pages 6–8 of the original Author Response). We would like to make it clear that we compare

[Figure AR1; Supplementary Figure 13 of the original manuscript]: (a) Near-field thermal radiative heat flux between bulk Ti media (black solid line), Ti/ MgF_2 multilayers (red dashed line), and Ti/dielectric ($\epsilon_d = 1.488$) multilayers (blue dotted line). The net radiative heat flux absorbed in each layer of metallo-dielectric receiver and corresponding cumulative heat flux for vacuum gaps of (b) 200 nm and (c) 400 nm.

Ti/MgF₂ multilayer with bulk Ti media, because Ti acts as the emitting layer (i.e., metal layer) but MgF₂ serves as the supporting layer (i.e., dielectric layer) in our metallo-dielectric multilayer. We are fully aware of the fact that the near-field thermal radiation between bulk MgF₂ is significant due to strong phonon absorption band of MgF₂. However, in our configuration, the role of MgF₂ is nothing but supporting layer (i.e., dielectric layer). As can be seen in Fig. AR1 (Supplementary Figure 13 of the original manuscript), if we replace the MgF₂ layers with lossless dielectric layers ($\epsilon_d = 1.488$), similar radiative heat flux can be achieved although lossless dielectric layers can neither absorb nor emit any radiation. Thus, comparison with bulk Ti media is quite fair because MgF₂ itself rarely contributes to the heat transfer in our configuration [refer to Fig. AR1b,c].

If we want to achieve the near-field enhancement over the bulk MgF₂ media, the same enhancement mechanism demonstrated in this work can be applied but MgF₂ needs to act as the emitting layer with another dielectric supporting layer. That is, the near-field thermal radiation between MgF₂/dielectric multilayers, where MgF₂ acts as the emitting layer (i.e., metal-like layer), can result in more heat transfer than the case of bulk MgF₂ media. For example, if we set MgF₂ as the emitting layer and introduce a lossless dielectric with $\epsilon = 1.35$ as the supporting layer, the near-field thermal radiation between MgF₂/dielectric multilayers (i.e., three unit cells of MgF₂/dielectric layers on a MgF₂ substrate) is larger than that between bulk MgF₂ media (21% enhancement) at vacuum gap of 200 nm (see Fig. AR2). Also, it can be clearly seen in Fig. AR2b,c that the enhancement is originated from the coupling of surface phonon polaritons supported at the multiple interfaces (MgF₂/dielectric interface) within the MgF₂/dielectric multilayers, which is the same enhancement mechanism that we have shown experimentally.

[Figure AR2; newly added as Supplementary Figure 14 of the revised manuscript]: (a) The net spectral heat flux between bulk MgF₂ media and between MgF₂/dielectric ($\epsilon_d = 1.35$) multilayers (three unit cells of MgF₂(200-nm-thick)/dielectric(200-nm-thick) on a MgF₂ substrate). The emitter and receiver temperatures are 400 K and 300 K, respectively. The vacuum gap width is 200 nm. (b) Computed $S_{\beta,\omega}^p$ between bulk MgF₂ media for vacuum gap distance of 200 nm and temperature of emitter and receiver conditions described in a. (c) As in b, but between MgF₂/dielectric multilayers.

In the revision, following the reviewer and editor’s suggestion, we have added the calculated near-field thermal radiation between bulk MgF_2 media as an inset of Fig. 4a [refer to the modified figure below]. Further, the discussion on the role of MgF_2 layer as the supporting layer in the considered Ti/MgF_2 MD structure and the fact that the same enhancement mechanism can be applied to the polar dielectrics are also added in the revised manuscript as well as in Supplementary Note 12.

[Figure AR3; modified as Figure 4a of the revised manuscript]: The calculated near-field radiative heat flux between bulk MgF_2 media is also plotted in the inset.

[REVIEWER 3]

In the revised version of the manuscript, the authors have fully addressed the comments of the three reviewers. The major claim of the paper (the enhancement of near-field thermal radiation with metallo-dielectric (MD) multilayers) has been more clearly supported by their additional experiment with the single MD structure. From my point of view, the paper is now suitable for publication in Nature Communications.

[AR]: We are grateful that the reviewer is now fully convinced by our response.

Reviewers' Comments:

Reviewer #1:

Remarks to the Author:

The authors explicitly compare their results of metallo-dielectric layers with those computed for MgF₂. I think this adequately addresses the concerns raised by Referee 2. I recommend publication.

Authors' Response

Manuscript ID: NCOMMS-18-17802B

Title: Tailoring near-field thermal radiation between metallo-dielectric multilayers using coupled surface plasmon polaritons

Authors: M. Lim, J. Song, S. S. Lee, and B. J. Lee

[REVIEWER 1]

The authors explicitly compare their results of metallo-dielectric layers with those computed for MgF_2 . I think this adequately addresses the concerns raised by Referee 2. I recommend publication.

[AR]: We thank the time and effort by all reviewers.